# Clinical Efficacy and Safety of Ultrasound-Guided Injection with Low-Molecular-Weight Peptides from Hydrolyzed Collagen in Patients with Partial Supraspinatus Tendon Tears: A Pilot Study

**DOI:** 10.3390/life14111351

**Published:** 2024-10-22

**Authors:** Luca Latini, Francesco Porta, Vincenzo Maccarrone, Davide Zompa, Edoardo Cipolletta, Riccardo Mashadi Mirza, Emilio Filippucci, Florentin Ananu Vreju

**Affiliations:** 1Centro Medico e Fisioterapia “Salute e Benessere”—Senigallia, 60019 Ancona, Italy; dr.lucalatini@gmail.com; 2Interdisciplinary Pain Medicine Unit, Rheumatology Section, Santa Maria Maddalena Hospital, Occhiobello, 45030 Rovigo, Italy; 3Rheumatology Unit, Department of Clinical and Molecular Sciences, Polytechnic University of Marche, 60131 Ancona, Italy; vincenzomaccarrone94@gmail.com (V.M.); edo.cipo@hotmail.it (E.C.); 4Radiology Unit, Riccione-Cattolica Hospital, 47923 Rimini, Italy; riccardo.mashadi@gmail.com; 5Department of Rheumatology, University of Medicine and Pharmacy of Craiova, 200349 Craiova, Romania; florentin.vreju@umfcv.ro

**Keywords:** shoulder pain, hydrolyzed collagen peptides, US-guided injections, ultrasonography, supraspinatus tendon tear

## Abstract

Background: This study evaluates the clinical efficacy and safety of two ultrasound (US)-guided injections of a 5 mg/1 mL low-molecular-weight peptide (LWP) solution derived from hydrolyzed bovine collagen in patients with supraspinatus partial tendon tears. Methods: A total of 21 patients with symptomatic partial tears of the supraspinatus tendon, detected by US, were consecutively enrolled and received one injection at a baseline visit (T0) and one after two weeks (T1). The primary outcome measure was the visual analogue scale (VAS) for pain. Secondary outcomes were the shoulder pain and disability index (SPADI) total score and the safety of LWP injections. Patients were examined at baseline (T0), at a week 2 follow-up visit (T1), and at a week 12 follow-up visit (T2). Results: A statistically significant improvement was found for both VAS pain and SPADI total scores, between T0 and T2 visits. US-guided injections were well tolerated and, apart from one patient with a progression of a tendon tear, no adverse events were recorded. Conclusions: Intratendinous tear US-guided injection therapy with an LWP solution was found to be safe and effective in improving both pain and shoulder function at a 12-week follow-up visit. The present pilot study should be considered the first step justifying a larger confirmatory investigation.

## 1. Introduction

Musculoskeletal pain is one of the most common sources of disability in the Western world, and the shoulder was found to be the painful site in about a third of the cases [1,2,3].

Persistent shoulder pain is a very common condition, involving both young and older patients, with the highest prevalence in manual workers and in athletes engaged in overhead sports. Such a condition often has a multifactorial pathogenesis and is associated with high social costs and patient burden [4,5]. 

Common causes of unilateral, persistent shoulder pain include bursitis, tendinopathy, rotator cuff (RC) tear, adhesive capsulitis, impingement syndrome, gleno-humeral joint osteoarthritis, and other causes of degenerative joint disease or trauma-related injuries, either in combination or as separate entities. In the general population, RC disorders account for 30–40% of all shoulder pain [6,7]. According to the Italian Society of Muscles, Ligaments, and Tendons (I.S.MU.L.T.) Guidelines, clinical and ultrasound (US) examinations are the two “starting points” to evaluate a patient with shoulder disorders [8]. 

In the last two decades, US has been increasingly used to provide a rapid and non-invasive visual confirmation of musculoskeletal pathology assessed by physical examination. This is especially true in complex anatomic sites such as the shoulder, where different structures can be involved, leading to a very similar clinical picture [9,10,11,12]. Moreover, the real-time imaging and clear visualization of the needle and target make US the most suitable tool for guiding injections [13,14,15]. Despite the fact that, in most cases, a non-guided procedure remains the standard in daily clinical practice, in our experience, US guidance is very important if not essential, especially when the injection efficacy depends on the exact placement of the tip of the needle and, thus, of the therapeutic agent in the target area. A partial rotator cuff tear is a small pathologic target that requires diagnostic imaging in order to be identified and topographically placed [16,17]. In fact, anatomic references are not enough to confirm the exact needle placement at the pathologic target level.

Although there is no unique validated protocol, conservative management is considered the first treatment approach for partial tendon tears (<1 cm in size). Initial non-surgical care can be safely undertaken in patients selected according to specific clinical and pathological characteristics: older age (>70 years old) with chronic tears, irreparable RC tears with irreversible pathological changes, including muscle atrophy and fatty infiltration together with superior humeral head migration and gleno-humeral arthropathy, and small (<1 cm in size) full-thickness tears or partial-thickness tears, regardless of age. In patients with the aforementioned characteristics, the conservative treatment appears to be a successful approach, with a positive functional outcome occurring in the range of between 40% and 60% of the cases [18]. Conversely, early surgical intervention should be considered in significant (larger than 1–1.5 cm) acute tears and in young patients with full-thickness tears who have a high risk for the development of irreparable RC changes [18,19,20]. 

In the last decade, the literature has underlined the role of hydrolyzed collagen as a therapeutic option in cases of osteoarthritis and other musculoskeletal disorders, including RC tendinopathy [21,22,23,24]. However, to the best of our knowledge, no studies have been published on the effectiveness of collagen low-molecular-weight peptide (LWP) injections in treating supraspinatus tendon tears. 

The rationale supporting a role for LWP injections in the treatment of tendon tears mainly relies on preliminary and limited, but encouraging, evidence from previous in vitro studies [25,26]. In particular, in the study by Randelli et al., an injectable medical compound containing collagen type I was able to induce an anabolic phenotype in cultured human tenocytes obtained from gluteal tendon fragments collected in patients who underwent total hip replacement [26].

The aim of this pilot study was to preliminarily investigate whether a treatment consisting of two US-guided injections of bovine-derived LWPs in patients with US-detected supraspinatus partial tendon tears could be clinically effective and safe. 

## 2. Materials and Methods

### 2.1. Study Design

The present study is a prospective, non-randomized, non-controlled pilot investigation involving consecutive patients with a symptomatic partial-thickness or small full-thickness supraspinatus tendon tear (maximum diameter < 1 cm in size) diagnosed by US. 

Patients were considered eligible for US-guided injections with LWPs if they met the following inclusion criteria: 18 years of age or older, persistent shoulder pain for at least one month, and US positive for partial-thickness or small full-thickness supraspinatus tendon tear (maximum diameter < 1 cm in size). 

The exclusion criteria included the following: the US detection of other relevant pathological changes of the shoulder [i.e., rotator cuff tears other than of the supraspinatus tendon, rotator cuff calcific tendinopathy, moderate/severe biceps tenosynovitis, moderate/severe gleno-humeral synovitis, and moderate/severe subdeltoid (SAD) bursitis], previous surgery or history of relevant trauma at the shoulder level, a known diagnosis of the rheumatic diseases crystal-related arthritis, inflammatory arthritis, fibromyalgia, or symptomatic cervical spine disease, and known drug hypersensitivity reactions.

In order to not influence any tissue-healing effects of LWPs, the only rescue medication allowed during the study was oral paracetamol at a maximum dose of 3 g/day. 

Patients were assessed three times: before the first injection of LWPs (baseline visit, T0), 2 weeks after the baseline visit (first follow-up visit, T1), and 12 weeks after the baseline visit (second follow-up visit, T2). During each visit, the following clinical data were recorded: visual analogue scale (VAS) for pain, shoulder pain and disability index (SPADI) [27]. 

Clinical data were recorded by two rheumatologists, not involved in the US examinations and US-guided injections. The study was conducted in accordance with the Helsinki Declaration and all patients gave their informed consent prior to enrolment. This research was conducted in accordance with the ethical standards and was approved by the Local Ethics Committee of Emergency County Hospital Craiova, Romania, with the number 22763/09.05.2023.

### 2.2. Ultrasound Examination

US examination was carried out to provide data to include or exclude patients from the study, to guide injections, and to assess shoulder pathology responsible for treatment failure.

Patients had their shoulders scanned with a MyLab Class C US system equipped with an 8–13 MHz linear probe or a MyLab X75 US system equipped with a 3–15 MHz linear probe (Esaote SpA, Genoa, Italy), according to the 2017 EULAR standardized procedures for US imaging [28]. All US examinations were performed by a rheumatologist blinded to clinical data, to detect US findings reported in the inclusion and exclusion criteria. US assessment was aimed at detecting a partial-thickness or small full-thickness supraspinatus tendon tear (maximum diameter < 1 cm in size) and excluding other relevant pathological changes of the shoulder. Supraspinatus tendon tear was defined as a hypoechoic or anechoic tendon defect with or without signs such as “cartilage interface” sign, focal flattening or concavity of the superficial tendon margin, cortical irregularity of the greater tuberosity, and/or mild synovial inflammatory signs (SAD bursitis and/or biceps tenosynovitis and/or gleno-humeral joint effusion) [29].

We considered a full-thickness tear of the supraspinatus tendon to be a lesion that involves the entire thickness of the tendon, while a partial-thickness tear implies an incomplete tear involving either the bursal, the intrasubstance, or the articular tendon aspect [30].

### 2.3. Ultrasound-Guided Injection

All patients underwent two US-guided injections with LWPs: at baseline and at the first follow-up visit. US-guided injections were performed according to a standardized protocol [13,16].

The injection was carried out using non-sterile gloves, non-sterile transducer’s cover, and chlorhexidine as a conductive medium between disinfected skin and the transducer’s cover. Further details on the asepsis and disinfection technique are described elsewhere [13].

All patients were asked to sit down in front of the physician, with the arm held in a neutral position, the elbow flexed at 90 degrees, and the forearm in a supinated position on the homolateral thigh. 

Such a protocol can be summarized in the following steps: -Operator having washed hands and wearing non-sterile gloves;-Preparation of the equipment: (1) collecting materials and (2) ensuring package seal integrity and checking expiration dates;-US probe setup: (1) cleaning of the probe using dry soft paper; (2) placing non-sterile gel on the probe footprint; and (3) putting a non-lubricated, non-sterile, rolled, transducer’s cover on the probe, and an alcohol–chlorhexidine solution was used as the conductive medium between disinfected skin and the sterile transducer’s cover;-Setup of the injection area: (1) skin area selection; (2) first disinfection of the entire injection field with povidone–iodine gauze of the target skin area using a circular motion, from the center to the edges (repeated 3 times); alcohol–chlorhexidine solution was used in cases of allergy to iodine; (3) application of cold non-sterile topical anesthetic (ethyl–chloride spray); and (4) second disinfection with povidone–iodine gauze of the target skin area using a circular motion (repeated 3 more times).

The patient was positioned on the examination bed with the injection area placed in a comfortable position to obtain full muscle relaxation. In some cases (e.g., patient without trunk control, previous episode of syncope), the intratendinous injection was performed with the patient lying down on the examination bed.

A rheumatologist used a free-hand and direct US-guided anterolateral in-plane approach to visualize the correct needle tip placement and the LWP injection within the tendon tear. A specific medical device was used (pre-prepared sterile disposable syringe with 5 mg/mL 1 mL of hydrolyzed bovine LWP solution, Tiss’You Srl RSM) to inject the LWPs into the intratendinous lesion. A 22-gauge needle was used. The correctness of the procedure and the spreading of the LWP solution was verified by real-time US assessment (Figure A1).

### 2.4. Outcome Measurements

The primary outcome measure was the VAS pain. The secondary outcomes were the SPADI total score and the safety of the LWP injections. The SPADI is a 13-item patient-reported index composed of 13 items assessing two main domains: pain (5 items) and disability (8 items) [30].

The VAS pain was chosen as the primary outcome measure as pain was one of the criteria to enter the study, while the SPADI was chosen as the secondary outcome measure because it is considered the preferred shoulder-specific questionnaire for assessing patients with painful shoulder in daily practice, according to an overall assessment of the main validity issues, including responsiveness to true change and feasibility [27]. 

The clinical efficacy of LWP injections was defined using the minimal clinically important difference values as an improvement of 1.37 cm for VAS pain and of 8 points for the SPADI [31,32].

### 2.5. Statistical Analysis

Qualitative variables were reported as absolute frequency and/or corresponding percentage, whereas quantitative variables, such as mean and standard deviation (SD) or median and interquartile range, were reported as appropriate. The Chi-Square test was used to compare qualitative variables and Student’s *t*-test and the Mann–Whitney U test were used to compare quantitative variables, as appropriate. Two-tailed *p*-values less than 0.05 were considered significant. Statistical analysis was performed using STATA (StataCorp, College Station, TX, USA) v.17.

## 3. Results

A total of 21 patients were enrolled. Their mean age was 64.2 (SD 10.8) years. Eleven patients (52.4%) were females. Only one patient did not complete the second follow-up visit.

A statistically significant improvement was found for both outcome measures of this study (i.e., VAS pain and SPADI total score), between the baseline (T0) and T2.

Table A1 and Figure A2 in Appendix A, report the VAS pain values and the SPADI total scores at the baseline and at follow-up visits.

With respect to the baseline values recorded at T0, an improvement in VAS pain value was found in 3 (14.3%) of the patients at the T1 follow-up and in 17 (81%) at the T2 follow-up visit. While a statistically significant difference in VAS pain values was not recorded between T0 and T1 (*p* = 0.07), a significant improvement was observed between T0 and T2 (*p* < 0.01) and between T1 and T2 (*p* < 0.01).

As regards the SPADI total score, an improvement was found in 13 (61.9%) of the patients at the T1 follow-up, and in 18 (85.7%) at the T2 follow-up visit. A statistically significant difference in the SPADI total score was found between T0 and T1 (*p* < 0.01) and between T0 and T2 (*p* < 0.01). Although a further improvement was recorded between T1 and T2, the difference was not statistically significant for the SPADI total score (*p* = 0.08).

Finally, US-guided injections were well tolerated in all patients. In only one patient, the VAS pain value did not improve (the VAS pain values at different visits were 71 mm at T0, 71 mm at T1, and 75 mm at T2), and at the T2 visit, US detected a progression of the supraspinatus tendon tear with a maximum diameter >1 cm in size.

## 4. Discussion

Tendon is a connective tissue component of the musculoskeletal system, involved in force transmission between muscle and bone. Its collagen structure is interlaced with numerous non-fibrillar proteins, which are essential to ensure the ability to support load with stability [7].

Tendon integrity is essential for musculoskeletal system function, and a tendon tear may entail a wide spectrum of disability. Tendon injury leads to an irreversible derangement of tendon texture, and to date, tendon structure “restitutio ad integrum” still represents a challenge for both clinical and research fields. Nevertheless, tendon function can be restored even in the absence of pre-tear structural integrity.

Tendon is mainly composed of type I collagen fibrils, displaying a parallel organization along the tendon axis. The tendon-specific spatial arrangement of type I collagen is essential for providing the mechanical properties necessary for tendon function [33]. Tenocytes, the main resident cells of the tendon, “sense” loads from the extracellular matrix (ECM), and, in turn, modulate the ECM [34]. Repetitive healthy loading, as in exercise, can promote remodeling in the tendon, leading to long-term structural and functional improvements. Tendon remodeling involves both the synthesis and the degradation of collagen with a net degradation that starts immediately after exercise and then shifts to a net synthesis. The observed breakdown of the ECM suggests that matrix metalloproteinases (MMPs) likely play a role in tendon adaptation [35]. Injured tendons do not fully restore the native ECM, leading to an altered biological and mechanical environment. Adult tendon healing is characterized by scar formation with disorganized tissue and reduced mechanical properties. Physiological exercise has been shown to increase the turnover of collagen I and promote an anabolic response [36]. The ability of type I collagen telopeptides to bind neighboring collagen molecules is known to be the initial event in fibrillogenesis [37].

In recent years, several biomaterials have been proposed to promote tendon healing [38], but to date, the evidence supporting their systematic use in patients with tendon tears is still scarce.

The use of medical devices based on collagen peptides, administrated orally or by intra-articular injections, was found effective in the management of knee osteoarthritis and in patients with rotator cuff pathologies, and in increasing muscle performance and in the prevention of soft tissue injuries in athletes [24,39,40,41,42,43]. Most of the studies have shown a statistically significant improvement in both shoulder pain and disability, with the achieved results being maintained up to a 6-month follow-up period [24,39,40,42].

Moreover, in vitro studies showed the efficacy of LWPs in promoting the biosynthesis of matrix molecules in tendons, ligaments, and cultured chondrocytes. Only a few studies investigated the possible role of collagen, but not LWPs, in the treatment of lateral epicondylitis and rotator cuff tears [44,45]. 

Our study is also based on the evidence that the stimulation of the extracellular matrix (ECM) synthesis is probably caused by specific Hyp-Pro-Gly-containing peptides with a molecular size <10 kDa [46,47,48].

To date, to the best of our knowledge, there are no studies in the literature focusing on the role of hydrolyzed collagen LWP injections in partial-thickness or small full-thickness rotator cuff tears.

To maintain the exploratory nature of this pilot study, we decided not to include randomized or a placebo-controlled group. In a previous work, a similar treatment protocol was shown to be effective when using US-guided soft tissue-adapted biocompatible hyaluronic acid injections in rotator cuff tears [16]. 

The present study showed how the use of a medical device based on LWPs derived from hydrolyzed bovine collagen was safe and effective in both reducing pain and improving shoulder function.

While at week 12, a statistically significant improvement was found for both VAS pain and SPADI total scores, patients reported a faster improvement in shoulder function than VAS pain values. 

The different timing of improvement of VAS pain and SPADI total scores, with VAS pain values decreasing more slowly than SPADI total scores, can be explained by the expected inflammatory effect within the tendon matrix caused by the LWP solution injections. 

Due to the exploratory nature of this pilot study, a definite result on LWP injections’ efficacy could not be achieved, and several drawbacks must be declared. First is the limited number of patients assessed in only one center and the lack of a control group. Second is the relatively short follow-up period (i.e., twelve weeks). Third, no other treatments were advised in the follow-up period except oral paracetamol at a maximum dose of 3 g/day in order to ascribe to LWP injection therapy any efficacy. Moreover, a more extensive clinical evaluation could be envisaged in addition to VAS pain and the SPADI. Finally, neither US nor MRI findings were used to assess LWP injections’ efficacy. In our study, US was used in the follow-up visits to detect shoulder pathology responsible for treatment failure, and a progression of tendon damage was found in only one patient.

Despite the declared limitations, the present study provided results justifying further research to confirm both the clinical efficacy and the safety profile of LWP injection therapy in a placebo-controlled multicentric investigation.

## 5. Conclusions

The obtained results preliminarily support the hypothesis that a series of two US-guided LWP injections is an effective and safe treatment option for patients with US-detected partial supraspinatus tendon tears. Although not definitive, our results are encouraging, and justify further studies in this research field to obtain more solid evidence on this topic.

## Data Availability

The data presented in this study are available on request from the corresponding author. The data are not publicly available due to General Data Protection Regulation (GDPR) reasons.

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
