# Peer review of "Clinical Efficacy and Safety of Ultrasound-Guided Injection with Low-Molecular-Weight Peptides from Hydrolyzed Collagen in Patients with Partial Supraspinatus Tendon Tears: A Pilot Study"

_life, 2024, doi:10.3390/life14111351_

Round 1

Reviewer 1 Report

Comments and Suggestions for Authors

Dear Authors,

I would like to congratulate you for the nice work submitted. Given the increasing attention to tendon pathologies, it is essential to provide more evidence on the use of new injectable drugs for their treatment.

The overall quality of the manuscript is good for its kind; however, the article needs some minor revisions before being suitable for publication.

INTRODUCTION

Line 39: not only in workers, but also in sportsmen engaged in overhead sports.

Line 42: please change "tendinitis" with "tendinopathy".

Line 63: you may report the difference, if any, between partial tears and "small" full-thickness tears.

Line 66: I would preferably report them as "pathological changes" instead of "changes" alone.

Line 75: you may cite a recent review on collagen for osteoarthritis:

Tarantino D, Mottola R, Palermi S, Sirico F, Corrado B, Gnasso R. Intra-Articular Collagen Injections for Osteoarthritis: A Narrative Review. Int J Environ Res Public Health. 2023 Mar 1;20(5):4390. doi: 10.3390/ijerph20054390. PMID: 36901400; PMCID: PMC10001647.

Lines 76-78: you should clarify that some studies on the use of infiltrative collagen (in general) have been already conducted for RC diseases, but not specificially on the use of infiltrative collagen peptides.

MATERIALS AND METHODS

Lines 126-128: I would delate the details about the other "pathological changes" since they were already reported before in the exclusion criteria.

Line 139: I would provide more details about the mentioned "standardized protocol".

Line 166: "were" considered.

RESULTS

Lines 198-199: please rephrase. Better state that "tendon function can be restored even in absence of pre-tear structural integrity"

Line 225: among the other studies on the use of injectable collagen for RC diseases, you should cite the following articles:

Corrado B, Bonini I, Chirico VA, Filippini E, Liguori L, Magliulo G, Mazzuoccolo G, Rosano N, Gisonni P. Ultrasound-guided collagen injections in the treatment of supraspinatus tendinopathy: a case series pilot study. J Biol Regul Homeost Agents. 2020 May-Jun;34(3 Suppl. 2):33-39. ADVANCES IN MUSCULOSKELETAL DISEASES AND INFECTIONS - SOTIMI 2019. PMID: 32856437.

Corrado B, Bonini I, Alessio Chirico V, Rosano N, Gisonni P. Use of injectable collagen in partial-thickness tears of the supraspinatus tendon: a case report. Oxf Med Case Reports. 2020 Nov 24;2020(11):omaa103. doi: 10.1093/omcr/omaa103. PMID: 33269086; PMCID: PMC7685015.

Kim JH, Kim DJ, Lee HJ, Kim BK, Kim YS. Atelocollagen Injection Improves Tendon Integrity in Partial-Thickness Rotator Cuff Tears: A Prospective Comparative Study. Orthop J Sports Med. 2020 Feb 21;8(2):2325967120904012. doi: 10.1177/2325967120904012. PMID: 32128319; PMCID: PMC7036510.

Chae SH, Won JY, Yoo JC. Clinical outcome of ultrasound-guided atelocollagen injection for patients with partial rotator cuff tear in an outpatient clinic: a preliminary study. Clin Shoulder Elb. 2020 May 25;23(2):80-85. doi: 10.5397/cise.2020.00066. PMID: 33330238; PMCID: PMC7714328.

Furthermore, despite the use of different kinds of collagen, a brief comparison with the outcomes of the above-mentioned studies could strengthen the evidence for the use of collagen for RC tears.

Line 134: please clarify that "STABHA" is a kind of injectable HA.

Comments on the Quality of English Language

Some sentences should be rephrased

Author Response

Reviewer 1

Dear Authors,

I would like to congratulate you for the nice work submitted. Given the increasing attention to tendon pathologies, it is essential to provide more evidence on the use of new injectable drugs for their treatment. The overall quality of the manuscript is good for its kind; however, the article needs some minor revisions before being suitable for publication.

Thank you to the reviewer for their positive feedback and insightful comments.

INTRODUCTION

Line 39: not only in workers, but also in sportsmen engaged in overhead sports.

            We fully agree with this suggestion and we added it in the text.

Line 42: please change "tendinitis" with "tendinopathy".

            We believe it is a good idea to change "tendinitis" with "tendinopathy" being the latter more inclusive. Please find the change in the text.

Line 63: you may report the difference, if any, between partial tears and "small" full-thickness tears.

            No differences. They are partial tears. Thank you for this point. The text has been amended accordingly.

Line 66: I would preferably report them as "pathological changes" instead of "changes" alone.

            We agree and “pathological” has been added in the text

Line 75: you may cite a recent review on collagen for osteoarthritis:

- Tarantino D, Mottola R, Palermi S, Sirico F, Corrado B, Gnasso R. Intra-Articular Collagen Injections for Osteoarthritis: A Narrative Review. Int J Environ Res Public Health. 2023 Mar 1;20(5):4390. doi: 10.3390/ijerph20054390. PMID: 36901400; PMCID: PMC10001647.

            The review has been added in the reference list (number 20).

Lines 76-78: you should clarify that some studies on the use of infiltrative collagen (in general) have been already conducted for RC diseases, but not specificially on the use of infiltrative collagen peptides.

            This point is well taken. We added this in the text and a reference supporting this aspect (number 24)

MATERIALS AND METHODS

Lines 126-128: I would delate the details about the other "pathological changes" since they were already reported before in the exclusion criteria.

            Thank you for pointing out this. Details in brackets describing other relevant pathological changes of the shoulder have been removed because redundant as already reported in the exclusion criteria.

Line 139: I would provide more details about the mentioned "standardized protocol".

            Thank you very much. We have now added more details about the standardised US-guided injection protocol.

Line 166: "were" considered.

            That is right: “was” has been replaced by “were”

RESULTS

Lines 198-199: please rephrase. Better state that "tendon function can be restored even in absence of pre-tear structural integrity"

            The text has been rephrased accordingly.

Line 225: among the other studies on the use of injectable collagen for RC diseases, you should cite the following articles:

- Corrado B, Bonini I, Chirico VA, Filippini E, Liguori L, Magliulo G, Mazzuoccolo G, Rosano N, Gisonni P. Ultrasound-guided collagen injections in the treatment of supraspinatus tendinopathy: a case series pilot study. J Biol Regul Homeost Agents. 2020 May-Jun;34(3 Suppl. 2):33-39. ADVANCES IN MUSCULOSKELETAL DISEASES AND INFECTIONS - SOTIMI 2019. PMID: 32856437.

- Corrado B, Bonini I, Alessio Chirico V, Rosano N, Gisonni P. Use of injectable collagen in partial-thickness tears of the supraspinatus tendon: a case report. Oxf Med Case Reports. 2020 Nov 24;2020(11):omaa103. doi: 10.1093/omcr/omaa103. PMID: 33269086; PMCID: PMC7685015.

- Kim JH, Kim DJ, Lee HJ, Kim BK, Kim YS. Atelocollagen Injection Improves Tendon Integrity in Partial-Thickness Rotator Cuff Tears: A Prospective Comparative Study. Orthop J Sports Med. 2020 Feb 21;8(2):2325967120904012. doi: 10.1177/2325967120904012. PMID: 32128319; PMCID: PMC7036510.

- Chae SH, Won JY, Yoo JC. Clinical outcome of ultrasound-guided atelocollagen injection for patients with partial rotator cuff tear in an outpatient clinic: a preliminary study. Clin Shoulder Elb. 2020 May 25;23(2):80-85. doi: 10.5397/cise.2020.00066. PMID: 33330238; PMCID: PMC7714328.

These studies have been added in the list of the references from number 24 to number 27.

Furthermore, despite the use of different kinds of collagen, a brief comparison with the outcomes of the above-mentioned studies could strengthen the evidence for the use of collagen for RC tears.

            This is a very good point. We have now added a sentence to discuss the outcomes of those studies (page 6).

Line 134: please clarify that "STABHA" is a kind of injectable HA

            Many thanks for pointing this out. Abbreviation (i.e. STABHA) was removed and all the words were reported (i.e. Soft Tissue Adapted Biocompatible Hyaluronic Acid).

Reviewer 2 Report

Comments and Suggestions for Authors

Dear colleagues,

Congratulations on this achievement.

For me, I find the topic very interesting and the title is well defined and matches the content. But I read and saw some problems.

I don't see any hypothesis of the work, where you started and where you want to go. - it must exist.

It would be good for your work to mention more citations to strengthen the rationale of your study, they are few.

There are some phrases in the text, which do not have a source of citation, there must be, and if there are many phrases from the same source, change it!

Add to the conclusions, in addition to confirming or refuting the hypothesis.

Few initial and final tests to validate from a physical point of view, joint mobility, clinical movement tests..., but I understood, and find it an excuse, that it is a pilot study.We definitely need objective evaluations in addition to VAS and SPADI.

Comments on the Quality of English Language

Dear colleagues,

Congratulations on this achievement.

For me, I find the topic very interesting and the title is well defined and matches the content. But I read and saw some problems.

I don't see any hypothesis of the work, where you started and where you want to go. - it must exist.

It would be good for your work to mention more citations to strengthen the rationale of your study, they are few.

There are some phrases in the text, which do not have a source of citation, there must be, and if there are many phrases from the same source, change it!

Add to the conclusions, in addition to confirming or refuting the hypothesis.

Few initial and final tests to validate from a physical point of view, joint mobility, clinical movement tests..., but I understood, and find it an excuse, that it is a pilot study.We definitely need objective evaluations in addition to VAS and SPADI.

Author Response

Reviewer 2

Dear colleagues,

Congratulations on this achievement.

For me, I find the topic very interesting and the title is well defined and matches the content. But I read and saw some problems.

We appreciate the reviewer’s comment on the achievement, their interest regarding the topic of our study, and their constructive suggestions.

I don't see any hypothesis of the work, where you started and where you want to go. - it must exist.

            Many thanks for raising this point. We amended the aim of the manuscript to formulate the hypothesis and we also changed the conclusion of the manuscript to answer whether the hypothesis was confirmed. 

It would be good for your work to mention more citations to strengthen the rationale of your study, they are few.

            We realized the in fact the number of citations regarding collagen injections in patients with rotator cuff tendinopathy could be increased, and we added more citations on this topic to strengthen the rationale of your study.

There are some phrases in the text, which do not have a source of citation, there must be, and if there are many phrases from the same source, change it!

We added a number of citations in the discussion and we hope the cover the gaps.

Add to the conclusions, in addition to confirming or refuting the hypothesis.

The conclusion of the manuscript was modified to answer to the question whether the results could confirm or refute the hypothesis.

Few initial and final tests to validate from a physical point of view, joint mobility, clinical movement tests..., but I understood, and find it an excuse, that it is a pilot study. We definitely need objective evaluations in addition to VAS and SPADI.

Thank you for bringing up this very important point. We do agree with you and we added a point in the limitations of the study.

Round 2

Reviewer 2 Report

Comments and Suggestions for Authors

congrats.

Comments on the Quality of English Language

congrats.